# Quantitative Study of Charge Distribution Variations on Silica–Nafion Composite Membranes under Hydration Using an Approximation Model

**DOI:** 10.3390/polym15102295

**Published:** 2023-05-12

**Authors:** Osung Kwon, Jaehyoung Park, Jihoon Lee

**Affiliations:** 1Faculty of Science, Tabula Rasa College, Keimyung University in Seongseo, Daegu 42601, Republic of Korea; 2Corporate Research Center, HI FUELCELL Co., Ltd., Daegu 41967, Republic of Korea

**Keywords:** proton exchange membrane, electrostatic force microscopy, numerical approximation model, proton conductivity, proton transport mechanism, ionic channel distribution, nafion composite, charge distribution

## Abstract

Understanding the ionic structure and charge transport on proton exchange membranes (PEMs) is crucial for their characterization and development. Electrostatic force microscopy (EFM) is one of the best tools for studying the ionic structure and charge transport on PEMs. In using EFM to study PEMs, an analytical approximation model is required for the interoperation of the EFM signal. In this study, we quantitatively analyzed recast Nafion and silica–Nafion composite membranes using the derived mathematical approximation model. The study was conducted in several steps. In the first step, the mathematical approximation model was derived using the principles of electromagnetism and EFM and the chemical structure of PEM. In the second step, the phase map and charge distribution map on the PEM were simultaneously derived using atomic force microscopy. In the final step, the charge distribution maps of the membranes were characterized using the model. There are several remarkable results in this study. First, the model was accurately derived as two independent terms. Each term shows the electrostatic force due to the induced charge of the dielectric surface and the free charge on the surface. Second, the local dielectric property and surface charge are numerically calculated on the membranes, and the calculation results are approximately valid compared with those in other studies.

## 1. Introduction

Proton exchange membranes are widely used in fuel cells and flow batteries [1,2]. Nafion, which is a sulfonated tetrafluoroethylene-based fluoropolymer–copolymer with conductive properties, is one such proton exchange membrane (PEM). Nafion and Nafion-based composite membranes have heterogeneous morphologies, such as the combination of a hydrophobic backbone with hydrophilic sulfonic acid groups [3]. Protons pass through a network through two diffusion mechanisms: the Grotthuss-type and vehicle-type mechanisms. In the Grotthuss mechanism, a proton diffuses into the hydrogen bonding network between a hydronium ion and water; in the vehicle-type mechanism, protic species (H_3_O^+^ or NH_4_^+^) attached to a proton diffuse into the membrane [4]. These two mechanisms simultaneously occur when protons are moved into the membrane. In wet conditions, the vehicle-type mechanism dominates; in contrast, the Grotthuss-type mechanism is dominant in dry conditions. The membrane proton conductivity is related to the distribution of the ionic channel network and water content on the membrane. Thus, understanding the proton conduction mechanism of a membrane is extremely complicated.

The most successfully commercialized proton exchange membrane is Nafion, which is developed by DuPont. Nafion shows good chemical characteristics and proton conductivity. The morphological structure of Nafion, which consists of a hydrophobic backbone and hydrophilic sulfonic acid groups, is an essential source of proton conductivity. Thus, its morphological structure has been studied by many groups since the 1980s [5,6,7]. For instance, Gierke et al. [5] conducted pioneering research for understanding the morphological structure of Nafion. They proposed their widely accepted cluster-network model with small-angle X-ray scattering and wide-angle X-ray scattering research. According to this model, an ionic cluster has an inverted micellar structure with spherical shape with a 2 nm radius and ionic clusters are scattered in a semicrystalline matrix. When the membrane is swollen, the ionic clusters are connected by channels with sizes of 1 nm, and an ionic channel network is created. When the water content in the membrane is increased, the ionic channel network becomes denser because of the increase in interconnection with ionic channels. Thus, an increased water content develops multiple proton pathways, enhancing the proton conductivity. Meanwhile, Klaus and Chen [6] proposed the cylindrical water channel model for Nafion through simulation studies based on previous scattering data. The cylindrical water channel model explains the morphology of Nafion as consisting of 2–5 nm cylindrical crystallites and forming cylindrical water channels with a radius of 2–3 nm in the polymer matrix. The crystallites support the mechanical durability of Nafion. The water channels, which are pathways of protons, increase in size as the volume of water increases on the Nafion.

Nafion has good proton conductivity; however, there are several characteristics that need improvement. First, Nafion has relatively weak mechanical characteristics, which is one of the reasons for the weakening of the reliability of the system [8,9]. Second, Nafion shows proper performance only under fully hydrated conditions. Thus, a humidifier is mandatory for the balance of the plant. This is not a problem for large-powered fuel cell systems, but it becomes a cause of reduced energy density for mid- or small-powered systems, such as portable devices. Thus, different types of PEMs have been developed by various research groups to solve the above problems.

One well-known method for enhancing PEM performance is the use of inorganic filler materials. Refs. [10,11,12,13,14,15,16,17,18] The purpose of using them is to increase the water containment of the membrane and ionic channel network density by using sulfonated filler materials. Graphene oxide was considered a good candidate for filler material because it has large specific area and functional group for sulfonization. Yoo et al. [10] synthesized Nafion composite membranes using different wt.% of Fe_3_O_4_–SGO. These composite membranes were fabricated using a facile solution casting method. From this previous study, the filler material helped increase the proton conductivity by 45% and reduce H_2_ crossover because the density of the –SO_3_H groups was increased by SGO and the hydrogen bonding networks between the –SO_3_H groups were prolonged by Fe_3_O_4_. Meanwhile, Shanmugam et al. [12] studied a polyoxometalate-coupled graphene oxide–Nafion membrane. They reported that the proton conductivities of Nafion 212 and the Nafion/mGO membrane were 6.5 and 9.2 mS cm^−1^, respectively.

TiO_2_ has strong hydrophilicity and proper characteristics for filler material that is used to enhance the water containment in the membrane. Ekani et al. [13] synthesized a Nafion/TiO_2_ composite membrane for high-temperature PEMFC using in situ sol–gel and casting. They reported that the proton conductivity of the Nafion/TiO_2_ composite membrane slightly decreased compared with that of pure Nafion even if the water uptake was increased by 50%. They concluded that the TiO_2_ particles blocked the proton movement. In contrast, Kim et al. [14] reported an enhanced performance of patterned mesoporous TiO_2_ microplates embedded in Nafion membranes. From their studies, the composite membrane showed a 35.2% maximum power density increase compared with that of pristine Nafion under 120 °C and relative humidity (RH) of 35% by increasing water retainment.

Various nanostructured silicas are widely considered filler materials. Mesoporous silica [15], nanotube-attached SiO_2_ [16], and nanoparticles [17] have been tested in this regard by many groups. For instance, Tang and Pen [18] developed a novel Nafion/SiO_2_ nanocomposite membrane by combining the self-assembly route and Nafion/SiO_2_ hybrids. Under 5 wt.% of Nafion-SiO_2_, the SiO_2_ nanoparticle showed good durability under hydration. The proton conductivity of the composite membrane under a 60% RH condition was 7.02 mS cm^−1^, which was roughly five times higher than the proton conductivity of Nafion 212.

In addition to the ones mentioned above, many research groups conducted studies on inorganic filler materials/Nafion composite membranes. Table 1 summarizes some of these studies. Most studies showed enhanced proton conductivity of the composite membrane under low humidification conditions.

The quantitative analysis of the mass, charged particles, and water transport near the membrane is widely studied by simulation and experimental methods. In simulation studies, the lattice Boltzmann method is used for the characterization of mass, charged particles, and the water transport phenomenon. Wang et al. [24] characterized the degradation mechanism of the gas diffusion layer (GDL) by the lattice Boltzmann method and Jeon et al. [25] characterized the water transport mechanism of the GDL. For experimental methods, X-ray scattering, atomic force microscopy (AFM), and electro transmission microscopy are widely used. Among them, AFM has great potential to quantitatively characterize charged particles and water transport on the membrane.

AFM can map not only morphology but also mechanical and electrical properties. The phase images from tapping-mode AFM are used to characterize the local stiffness of the sample surface. The vibrating cantilever near resonance frequency and the sample surface create a damping system. The frequency of the system varies with energy dissipation between the tip-attached bottom of the cantilever and the sample surface. Using the measured frequency shift, the local stiffness on the sample surface can be mapped at nanometer scale. Thus, it is a useful tool to characterize mechanically heterogeneous materials [26,27].

EFM is an extended mode of AFM. It can measure the morphology and electrical structure of a membrane surface. In many membrane studies, AFM is used to characterize the surface charge distribution and dielectric constant of locally charged materials [28,29,30]. EFM can measure long- and short-range forces simultaneously, such as van der Waals and electrostatic forces, using tip vibration change due to each interaction. From the short-range force, topography, which can map the surface morphology, is measured; from the long-range force, surface charge distributions can be mapped simultaneously. In EFM measurement, the detailed surface charge distribution can be observed using the phase value, which is proportional to the frequency shift of a vibrating tip due to the electrostatic interaction between a tip and the sample surface. However, understanding the charge value on specific locations is difficult because it is a result of unknown net forces from different types of charges. Thus, a mathematical approximation model is required based on the principles of EFM to understand the EFM signal.

Many analytical models have been proposed through understanding the principles of EFM [31,32,33]. For instance, Mélin et al. [31] introduced an analytical model assuming EFM as a nanosized parallel plate capacitor for estimating the stored charge value on a surface. They assumed that the force gradient was due to the dipole–dipole interaction between a tip and the stored charge, and they calculated the stored charge numerically. In another study, Han et al. [32] studied low-density polyethylene (LDPE) on injected charges. They interpreted the phase value of EFM from LDPE assuming a tip and the LDPE surface consisting of nanosized capacitors. The numerical model of charge distribution on the surface showed that the phase value from EFM was a result of a net force due to the local charge between a tip and the sample surface. In this previous study, local charges were proportional to the phase value. Meanwhile, Shen et al. [33] proposed a remarkable approximation model for the quantitative interpretation of the phase value from EFM based on EFM measurements of the monolayer graphene oxide sheet. From this model, they also assumed a nanocapacitor as a tip and the sample surface. A capacitive force was applied between the tip and the sample surface, and the force gradient of the difference in the dielectric constant of graphene and mica appeared as a phase value variation.

A previous study on a silica–Nafion composite membrane [34] showed superior proton conductivity enhancement under insufficient hydration conditions, as shown in Table 2. The study focused on understanding proton conductivity enhancement based on the numerical approximation model and electrostatic force microscopy (EFM) measurement. However, the model was oversimplified and did not provide numerical information. Thus, a nonoversimplified model that reflects physical phenomena between a tip and the membrane surface is required for a relatively accurate analysis of the morphology of the membrane.

This study was conducted to achieve two purposes. First, we aim to derive an approximation model for the quantitative analysis of PEMs. The approximation model for the interpretation of the EFM signal was derived based on the principle of electromagnetism, EFM, and detailed environmental conditions of the PEMs. Typically, a water meniscus is created between a tip and the sample surface because of capillary force [35]. In addition, hydrophilic reinforced membranes, such as the silica–Nafion composite membrane, have comparably thick water layers on the membrane surface. If the tip and the membrane surface are assumed as parallel capacitors, two different dielectric layers exist: the water and the membrane itself. These double-layered dielectric materials are considered in the development of the advanced numerical approximation model. The second purpose of the study is to verify the approximation model using a developed proton exchange membrane. Here, recast Nafion and silica–Nafion composite membranes were prepared to verify the approximation model. These membranes were measured using EFM and analyzed using the approximation model. 

Local dielectric properties at the nanoscale depend on the structure and morphology of the medium. Additionally, ionic structure, which is connected with the phase separation of PEMs, is crucial for understanding the proton conductivity variation of PEMs. The novel quantitative approximation model developed based on EFM measurement can numerically analyze the local dielectric properties and surface charge distribution at the nanoscale, providing quantitative information about ionic interaction near the surface and morphology. Thus, this model can help in providing a detailed understanding of PEMs and analyzing novel PEMs.

## 2. Experimental Setup

This study was conducted in several steps. The first step was modeling based on electromagnetic dynamics and the basic principles of EFM. The second step was measuring recast Nafion and silica–Nafion composite membranes. Finally, the surface charge distribution was characterized using the mathematical approximation model.

EFM is a powerful tool for understanding surface charge distributions. Typically, EFM works in noncontact mode. While a conductive tip, which vibrates near the resonance frequency, is scanning, a DC bias voltage is applied between the tip and the sample surface. Then, electrostatic and van der Waals forces are applied between the conductive tip and the sample surface. The morphology of the surface is mapped from the van der Waals force, whereas the surface charge distribution is simultaneously mapped using the electrostatic force. If the force between the tip and the sample surface is an attractive regime, the resonance frequency becomes low. Then, the frequency shift has a negative value. The phase signal of EFM is the product of the amplitude of the tip and the frequency shift. Thus, the phase value is negative under an attractive force, as shown in Figure 1a. When the force between the tip and the sample surface is repulsive, the resonance frequency is raised and the phase value is positive, as shown in Figure 1a. For strong dielectric materials, the electrostatic interaction is always an attractive force, as shown in Figure 1b,c. When charged particles, which are of opposite polarity compared with the electrical polarity on the sample surface, are placed on the sample surface and the charged particles have a sufficiently strong electrical charge, a repulsive force can occur to the tip. By this principle, the surcharge distribution can be mapped using EFM.

In this study, recast Nafion, which is exposed under ambient conditions, was prepared for measuring and analyzing surface charge distributions using the mathematical approximation model. In a previous study, the recast Nafion was quantitatively analyzed under dried and fully hydrated conditions [36]. The purpose of using recast Nafion in this study was to verify the reliability of the numerical approximation model. The results were compared with other research groups’ studies [37,38,39]. Additionally, silica–Nafion composite membranes were prepared under different conditions. One silica–Nafion composite membrane was prepared under ambient conditions, and it was called a composite membrane under room humidity. Another silica–Nafion composite membrane was fully hydrated, and it was called a composite membrane under a fully hydrated condition. For this, a composite membrane was soaked in deionized water overnight. The membranes were taken out of a vessel before measurement. The measurement and analysis were accurately and systematically conducted in a few steps. First, 10 μm × 10 μm membranes were scanned at a scan frequency of 1 Hz without applying a sample bias voltage. In this step, topography and phase maps were created to characterize the morphology and local mechanical property of the membrane surface. Second, all 10 μm × 10 μm membranes were scanned by applying a positive 2 V bias voltage to create a surface charge distribution map. Finally, from the electrostatic phase shift of the membrane surfaces, the phase shift variations over the membranes were characterized using the mathematical approximation model.

## 3. Modeling

A bias voltage was applied between a tip and the sample surface to measure the electrical charge distribution on the sample surface. Additionally, the surface charge distribution on the proton exchange membrane was measured using the same method. However, characterizing surface charge distributions compared with other conducting surfaces is not easy. For analyzing charge distribution measurements, the existence of a capacitance force between a tip and the sample surface is typically assumed and modeled. However, modeling PEMs is more complicated. In a previous study, modeling was conducted by assuming a nanocapacitor with a dielectric slab, which comprises a tip/membrane/sample holder. However, this was an oversimplified model because the geometry of a conical tip was assumed as a flat surface and the noncontact tip and the sample surface were assumed as in contact. The actual environment between a tip and the sample surface is more complicated, as shown in Figure 2. A water layer exists between a tip and the sample surface, and this is widely accepted by other studies [40].

A schematic of a nanocapacitor due to a tip and sample surface shows a capacitor with double slabs, as in Figure 2. Using this schematic, a new analytical model was derived. In this model, a tip, the water layer, the membrane, and a conducting sample holder consisted of an electrode, the first dielectric slab, the second dielectric slab, and an electrode. In this modeling, the tip was assumed as a flat disk with a radius *R_tip_*.

In a proton exchange membrane, the electrostatic force between a tip and the sample surface is the net force of the capacitive and Coulombic forces due to free charges (*Q_free_*) on the surface. The capacitive force *F_C_* is proportional to the first derivative of the capacitance of a nanosized capacitor, such as a tip/water layer/membrane/sample holder. Equation (1) shows the capacitive force, where *C* and *V* represent the capacitance and bias voltage, respectively.
(1)$$F_{c}=\frac{1}{2}\frac{\partial C}{\partial z}V^{2}$$

When two different dielectric materials are placed between two electrodes of a capacitor, the capacitance is derived as
(2)$$C=\frac{\varepsilon_{0}S}{\frac{z}{\varepsilon_{1}}+\frac{t}{\varepsilon_{2}}}$$
where *z*, *t*, *ε*_1_, *S*, and *ε*_2_ are the thickness of the first dielectric slab, the thickness of the second dielectric slab, the dielectric constant of the first material, the area of the capacitor, and the dielectric constant of the second material, respectively, as shown in Figure 2. Equation (2) is substituted to Equation (1), and the capacitive force is expressed as Equation (3), where *Fc* and *ε_0_* are capacitance force and vacuum permittivity.
(3)$$F_{c}=-\frac{\varepsilon_{0}S}{\varepsilon_{1}{\left(\frac{z}{\varepsilon_{1}}+\frac{t}{\varepsilon_{2}}\right)}^{2}}V^{2}$$

The electrostatic force due to free charges (*Q_free_*) on the surface *F_f_* can be approximated using the Coulombic law.
(4)$$F_{f}=\frac{1}{4\pi \varepsilon_{0}}\frac{Q_{free}Q_{tip}}{z^{2}}$$
where *Q_free_* and *Q_tip_* are the free charge and the tip charge, respectively. The basic principle of capacitance with a dielectric material is used to find the expression of *Q_tip_*. The electric field *E* under a dielectric slab is
(5)$$E=\frac{E_{0}}{k}≅\frac{1}{k}\frac{V}{d}$$
where *E*, *E*_0_, *k*, *V*, and *d* are the electric field with a dielectric slab, the electric field without a dielectric slab, the dielectric constant, and the applied bias voltage, respectively. The electric field without a dielectric slab is the surface charge density *σ* over vacuum permittivity *ε*_0_. Thus, Equation (5) becomes
(6)$$\sigma =\frac{\varepsilon_{0}V}{d}$$
and
(7)$$\sigma =\frac{Q_{tip}}{S}$$
where *S* is the surface area of the tip. The charge on a tip (*Q_tip_*) can be
(8)$$Q_{tip}=\frac{\varepsilon_{0}SV}{d}$$

Equation (8) is substituted to Equation (4), and the electrostatic force due to free charges is
(9)$$F_{f}=\frac{1}{4\pi }\frac{Q_{free}S}{{dz}^{2}}V$$

The net force is expressed as follows:(10)$$F=F_{c}+F_{f}=-\frac{\varepsilon_{0}S}{\varepsilon_{1}{\left(\frac{z}{\varepsilon_{1}}+\frac{t}{\varepsilon_{2}}\right)}^{2}}V^{2}+\frac{1}{4\pi }\frac{Q_{free}S}{{dz}^{2}}V$$

The frequency shift value of a tip from the EFM measurement is proportional to the force gradient.
(11)$$\frac{\partial F}{\partial z}=\frac{2\varepsilon_{0}S}{\varepsilon_{1}^{2}{\left(\frac{z}{\varepsilon_{1}}+\frac{t}{\varepsilon_{2}}\right)}^{3}}V^{2}-\frac{1}{2\pi }\frac{Q_{free}S}{{dz}^{3}}V$$

The frequency shift value is the same as follows:(12)$$∆f≅-\frac{F\prime }{2K}f_{0}=-\frac{\varepsilon_{0}S}{{K\varepsilon }_{1}^{2}{\left(\frac{z}{\varepsilon_{1}}+\frac{t}{\varepsilon_{2}}\right)}^{3}}f_{0}V^{2}+\frac{1}{4\pi }\frac{Q_{free}S}{K{dz}^{3}}f_{0}V$$

*K* and *f*_0_ in Equation (12) are the spring constant of a cantilever and resonance frequency, respectively. 

The phase shift value, which is directly connected with the surface charge distribution, is the product of the amplitude *A* and the frequency shift. Thus, the phase shift value is expressed as Equation (13).
(13)$$∆\emptyset ≅A∆f=-\frac{\varepsilon_{0}SA}{{K\varepsilon }_{1}^{2}{\left(\frac{z}{\varepsilon_{1}}+\frac{t}{\varepsilon_{2}}\right)}^{3}}f_{0}V^{2}+\frac{1}{4\pi }\frac{Q_{free}SA}{Kdz^{3}}f_{0}V$$

The mathematical approximation model is expressed in two terms, as shown in Equation (13). The first term of the model is related to the induced surface charge, and it is dominant where the surface is a dielectric material with weak free charges. The second term of the model reflects the electrostatic interaction due to free charges on the surface. Under the local existence of strong free charges, the second term of the model is dominant. This model can be used not only for PEMs but also for heterogeneous materials with locally conductive areas. However, for the proper use of the model, understanding the electrostatic characteristics of surfaces should be prioritized.

## 4. Experimental Results

The topography and line profile of recast Nafion and silica–Nafion composite membranes under room humidity and a fully hydrated condition are shown in Figure 3. The colors on the topography image represent various heights on the surface. When the color is brighter, the height on the surface is high; when the color is darker, the height on the surface is low. Each topography does not show any ordered morphological structure. Recast Nafion under room humidity shows a smooth surface without any remarkable structures, as shown in Figure 3a. Additionally, the surface of the membrane is clean, and dirt or extrusion is not observed. In the topography of the silica–Nafion composite membrane under room humidity, most areas on the membrane surface show relatively rougher surface variations, and the brightest areas (mid-left) are observed on the topography, as shown in Figure 3b. The bright areas show several hundred height variations, which might be due to the coagulation of silica on the membrane surface. The evidence of silica coagulation is provided in the explanation of the phase-mode AFM. The topography of silica–Nafion under a fully hydrated condition shows more rough variations, which are due to the increased water content on the membrane, as shown in Figure 3c. Additionally, several extruded areas a few hundred nanometers higher than the circumference are observed, which might be coagulated areas.

For quantitative analysis, the root mean square (RMS) roughness was calculated on the basis of the membrane mean surface height variation. The RMS roughness values of dried recast Nafion and the silica–Nafion composite membrane under room humidity and a fully hydrated condition were 1.73, 6.62, and 47.0 nm, respectively. The recast Nafion had an extremely smooth surface compared with the silica–Nafion composite membrane. The silica–Nafion composite membrane under room humidity had a three times rougher surface compared with that of the recast Nafion. In the fully hydrated condition, the RMS roughness was eight times larger than that under room humidity. This result implies that the morphology of the surface of the silica–Nafion composite membrane became rougher and that a coagulated area was created compared with the recast Nafion.

The phase mode of AFM can measure mechanical properties on the membrane surface, such as elasticity, viscosity, and viscoelasticity, using a vibrating tip. When a vibrating tip near its resonance frequency approaches the membrane surface, there exists an interaction between the tip and the membrane surface due to short-range force, such as the van der Waals force. Through this interaction, the morphology of the membrane surface can be mapped. Additionally, the frequency of the tip can be shifted by the mechanical properties of the membrane. When a scanned area on the membrane surface is an elastic region, the phase shift is small. When the tip is located on an energy-dissipative region, such as viscous or viscoelastic materials, the frequency variation is increased. The phase lag can be calculated as the product of the frequency shift and amplitude. As the phase lag is proportional to the amount of frequency shift, a high-phase-lagged area refers to an energy-dissipative area.

Figure 4 shows a phase image of the recast Nafion and the silica–Nafion composite membrane under room humidity and a fully hydrated condition. The phase image of recast Nafion agrees well with its topography, and the phase value varies from −7° to −15°. A remarkable phase value change was not observed on the membrane surface. This means that the mechanical property on the surface is uniform. For quantitative analysis, a histogram from the phase map of recast Nafion was extracted, as shown in Figure 4d. The peak value and full width at half maximum (FWHM) were −11° and 2°, respectively. From the relatively small peak value and FWHM, the energy dissipation of recast Nafion is small and the energy dissipation variation is uniform. Meanwhile, the images of silica–Nafion composite membranes do not agree with each topography. For the silica–Nafion composite membrane under room humidity, the phase values varied from −10° to −30°. Some areas, which show brighter colors than those of other areas, indicate low phase values, and these areas are less energy dissipative than other areas. The negative phase value shows the interaction between a tip and the membrane surface as a repulsive force. For quantitative analysis, the peak value and FWHM were calculated from the histogram, as shown in Figure 4e, which were −20° and 3.2°, respectively. The peak value of the membrane was two times larger than that of the recast Nafion, which showed that the energy dissipation of the membrane surface was much higher than that of the recast Nafion. The phase value difference, which was concluded from the extremely small FWHM of the histogram, provides conjecture of the monotonous mechanical properties of the surface. From these results, the mechanical properties of the extruded areas of morphology did not differ from those of other areas, but the elasticity was slightly higher than those of other areas. Thus, the extruded area might be due to the coagulation of silica.

Figure 4c shows a phase image of the silica–Nafion composite membrane under a fully hydrated condition. The phase value varied from −100° to 100°. The bright and dark colors on the image indicate positive and negative phase values, respectively. The area with a positive phase value on the membrane surface shows that a tip is in contact while scanning because of the existing relatively strong attraction force between the tip and the surface. The mean phase value of the membrane surface was 70°, indicating high energy dissipation. This high energy dissipation implies that the surface is viscous or viscoelastic. More viscous-conditioned membrane surfaces point to the existence of a water layer on the membrane surface. Thus, the attraction force between the tip and the surface was due to increased capillary force by the existing water layer on the membrane surface [35]. From the related histogram, the peak value and FWHM were 70° and 13°, respectively, as shown in Figure 4f.

Directly comparing silica–Nafion composite membranes under different conditions is difficult because the phase values due to the tip and surface interaction are opposite, such as the negative phase value from the silica–Nafion composite membrane under room humidity and the positive phase value from the silica–Nafion composite membrane under a fully hydrated condition. From Figure 4c, protrusions are not observed from the entire membrane surface. The FWHM is relatively large compared with those of recast Nafion and the silica–Nafion composite membrane under room humidity. This result implies that the dispersion of the phase value, which relates to energy dissipation, is relatively large and the energy dissipation on the surface is not uniform. Additionally, the protrusions observed from the topography might be a coagulation of silica incoherent with the topography.

Figure 5 shows EFM images of the recast Nafion and silica–Nafion composite membrane under room humidity and a fully hydrated condition by applying 2 V between the tip and the membrane surface. Each pixel of the images represents the electrostatic interaction between the tip and the membrane surface at the bottom of the tip. The colors of the pixels are the intensities of the phase values. When the color is bright, the phase value is higher. Figure 5a,d show the EFM image and the related histogram of the recast Nafion composite membrane. The phase value of the surface shows a uniform color, implying that the electrostatic interaction between the tip and the surface is not drastically changed like the phase map and topography. The phase value roughly varied from −90° to −120°. All phase values on the membrane surface were negative. This is due to the application of an attractive force between the tip and the membrane surface. The mean phase value and FWHM were −104° and 7°, respectively. Figure 5b,e show the EFM image and the related histogram of the silica–Nafion composite membrane under room humidity. The image does not show any coherence with the topography and the phase image, which were scanned simultaneously. The phase value of the entire surface is located in the negative value. This also shows that the electrostatic interaction between the tip and the surface is an attractive force. The mean phase of the surface is −96°, and the FWHM is 1.3°. The mean phase value and FWHM of the silica–Nafion composite membrane under room humidity were reduced compared with those of recast Nafion. A direct interpretation using the phase map from EFM is difficult; however, by inference, this reduction might be related to the changes in surface charge distribution due to the water contents on the membrane surface. A more detailed characterization is conducted using the mathematical approximation method in the next section.

Figure 5c,f show the EFM image and the related histogram of the silica–Nafion composite membrane under a fully hydrated condition. The figure shows a drastic phase value change compared with that of the recast Nafion and the silica–Nafion composite membrane under room humidity. The image shows a clear phase separation on the surface, such as bright and dark regions. The bright-colored region was dominantly placed on the membrane surface. The electrostatic interactions between the tip and the surface of the bright- and dark-colored regions were repulsive and attractive forces, respectively. The bright color, which was due to repulsive force, shows that the tip and the membrane surface have electrical charges with the same polarity. The phase separation indicates that the electrical charge distribution on the membrane surface is locally discontinuous because the electrostatic force is discontinuously charged, such as the attractive force (the dark area on the membrane surface) and the repulsive force (the bright area on the membrane surface). The phase separation on the surface is more clearly seen in the histogram shown in Figure 5f. The peak value and FWHM of the bright region were 74° and 13°, respectively. Meanwhile, the peak value and FWHM of the dark region were −118° and 13°, respectively.

## 5. Analysis

The surface charge distribution of each membrane was characterized using the derived mathematical model. Each phase value can be analyzed using Equation (13), which consists of two terms. The first term represents the electrostatic force due to induced charges on the membrane surface. In dielectric materials, the first term is dominantly related to the measured phase value. The second term represents the electrostatic force due to free surface charges. When free surface charges are dominant, the second term is strongly affected by the phase value. Typically, the phase value of an EFM image is the sum of both terms. In recast Nafion, the first term might be dominant because the water content of the membrane is dependent on the insufficient water layer on the membrane, as shown in Figure 4d [40]. From the phase image, the energy dissipation on the surface is the smallest value among those of various membranes. This implies that a water layer between the tip and the membrane surface does not exist or is thin. Thus, ionic clusters in the matrix did not create ionic domains on the membrane surface. A tip is coated with Pt. When a water layer exists between the tip and the membrane surface, electrolysis might occur. In this case, electrolysis is weak because of the insufficient water layer between the tip and the membrane surface. Thus, in the membrane surface, the surface charge can be rationally attributed to the rare existence of free charges on the membrane surface. From this assumption, the first term from Equation (13) affects the dominant phase value, whereas the effect of the second term is very weak. Then, the free charge density *σ* is assumed zero.
$$\frac{1}{4\pi }\frac{Q_{free}SA}{{Kdz}^{3}}f_{0}V≅0$$

In this case, Equation (13) is approximated as
(14)$$∆\emptyset_{Nafion}≅-\frac{\varepsilon_{0}AS}{{K\varepsilon }_{1}^{2}{\left(\frac{z}{\varepsilon_{1}}+\frac{t}{\varepsilon_{2}}\right)}^{3}}f_{0}V^{2}$$

A local dielectric constant of recast Nafion was calculated to prove the reliability of Equation (13). Table 3 shows related parameters and corresponding values. Most values show experimental conditions and manufacturing information. The radius of the tip for calculating area *S* was used for the blind reconstruction method [41].

From the calculations, the local dielectric constant of recast Nafion is 14. This value is roughly similar to those of other studies [37,38,39], as shown in Table 4. Thus, this validates the characterization of local dielectric properties on the membrane.

The silica–Nafion composite membrane under room humidity can also be analyzed using the proposed model; however, the charge distribution on the surface is not the same as that of the recast Nafion. The phase value of the silica–Nafion composite membrane under room humidity was two times larger than that of the recast Nafion from the results of the phase-mode AFM. This implies that the energy dissipation of the surface is increased compared with that for the recast Nafion. The increased energy dissipation shows the enhancement of the viscosity on the surface. Then, it can be concluded that the water layer on the surface is increased, as shown in Figure 6. When a bias voltage is applied between the tip and the membrane surface, electrolysis occurs. Near the tip coated with Pt, protons and electrons are generated. Thus, the protons become free charges on the surface. The net interaction between the tip and the membrane surface is the sum of induced charges with negative polarity and free charges (i.e., protons). Thus, both terms in Equation (13) affected the phase value. The net electrostatic force between the tip and the membrane surface might be reduced because induced charges, which have negative polarity under the application of a negative sample bias, and free charges, which are protons, have opposite electric polarities. 

The mean phase value from the silica–Nafion composite membrane under room humidity was −96, which was less than the mean phase value of the recast Nafion. The phase value of the entire surface is negative, showing that the tip and the membrane surface have an attractive force. The mean phase value was −96, which was due to the net electrostatic force induced by the surface charge and free charges. The first term from Equation (13), which relates to the induced surface charge, might be approximated using the phase value of the recast Nafion. Then, Equation (13) becomes
(15)$$∆\emptyset ≅∆\emptyset_{Nafion}+\frac{1}{4\pi }\frac{Q_{free}SA}{Kdz^{3}}f_{0}V$$

Using this equation, the surface free charge can be calculated. Table 5 shows related parameters and corresponding values. From the calculation, the free charge under the tip is approximately 1.0 × 10^−9^ C. However, verification of the approximation result calculated from the model is difficult. The current sensing atomic force scope (CSAFM) can measure the nanoscopic current distribution of the Nafion. Figure 7 shows the result obtained using CSAFM, revealing that the current distribution on the surface is roughly a few nA. Typically, current is defined as the deviation of the total electrical charge value by time, dc/dt. Considering a scan rate of 1 Hz, the charge that moved through the ionic domain was ~0.03 × 10^−9^ C. The tip radius of the image was assumed to be a few tens of nanometers from the blind reconstruction method, and it was 10 times smaller than the tip radius used for measuring EFM. Moreover, the CSAFM only measures moving electrons from electrolysis. Thus, electrical charges affecting electrostatic force might be more than 10 times larger. Thus, the calculated value from the approximation model might be reasonable. 

The silica–Nafion composite membrane under a fully hydrated condition shows a drastic phase value difference compared with that of the silica–Nafion under room humidity. From the histogram, a strong positive peak is observed near 74°, whereas a weak negative peak is observed near −118, as shown in Figure 5f. This is due to the ionic domain creation caused by the fully hydrated condition and the increased water layer on the membrane surface, as shown in Figure 6b. Additionally, hydrolysis might have affected the phase separation. Hydration is induced by the increased ionic domain and the creation of a network among domains. Then, the watery area on the surface is increased and proton from the ionic channel and hydrolysis might be widely located, as shown in Figure 6b. Thus, the repulsive force is strongly affected between the tip and the membrane surface. In the negative-phase-value area, the negative charge density was high because of the induced surface charge and negative charge from the ionic cluster, as shown in Figure 6b. Thus, a strong attractive force was applied between the tip and the membrane surface. Free charges then dominated the positive-phase-value area, whereas induced charges dominated the negative-phase-value area.

The free charges, which were mainly protons, were directly connected to the size of the ionic domain. The free charge density may be calculated using the mathematical approximation model. In the positive-phase area, the second term of Equation (13) dominated. The positive-value area was watery, and the negatively induced surface was located around or beneath the watery area. Thus, the first term might be extremely small and can be neglected. Thus, Equation (13) can be expressed as
$$∆\emptyset ≅\frac{1}{4\pi }\frac{Q_{free}SA}{Kdz^{3}}f_{0}V$$

Using the phase value, the result is roughly 9.6 × 10^−9^ C. The surface free charge value of the silica–Nafion composite membrane in the repulsive-force-dominant area under a fully hydrated condition was 90 times larger than that of the silica–Nafion composite membrane under room humidity. If considering all of the scanned area, the surface charge value might be less than 90 times. This result is not direct evidence of an increased ionic domain; however, it can be indirect evidence of the enlargement of the ionic domain by calculating the surface free charge because a large ionic domain can contain relatively many protons. Additionally, this result shows similarities with the proton conductivity under different hydration conditions. The proton conductivity under the fully hydrated condition was 60–70 times larger than that under room humidity in many studies [36]. These results agree well with the result from the surface charge calculation based on the model.

## 6. Conclusions

In this study, an approximation model was used to characterize the surface charge of PEMs. The model was based on simple electromagnetic dynamics and the basic principles of EFM. This model was derived under two assumptions. First, a tip and the sample holder were assumed as nanosized capacitors. Second, between the capacitor, two slabs (the membrane and a water layer) were sandwiched. The model consists of two terms: the related induced surface charge and the free charge on the surface. The local dielectric property can be approximately calculated from the model. Additionally, the surface charge distribution can be roughly calculated. According to the model, recast Nafion and silica–Nafion composite membranes can be characterized. The mean dielectric property of 10 μm × 10 μm was approximated based on the model, and the result agreed well with those of other groups. Using the model and the electrochemical interaction on the recast Nafion composite membrane, surface free charges, which are related to the ionic domain, were calculated for the silica–Nafion composite membranes under room humidity and the fully hydrated condition. Comparing the results for the two conditions, the free charge from the fully hydrated condition was 90 times larger than that under the room humidity condition. This result shows a similar trend with increasing proton conductivity as those in other studies. Thus, the model can be utilized for approximating local dielectric properties and charges.

## Figures and Tables

**Figure 1 polymers-15-02295-f001:**
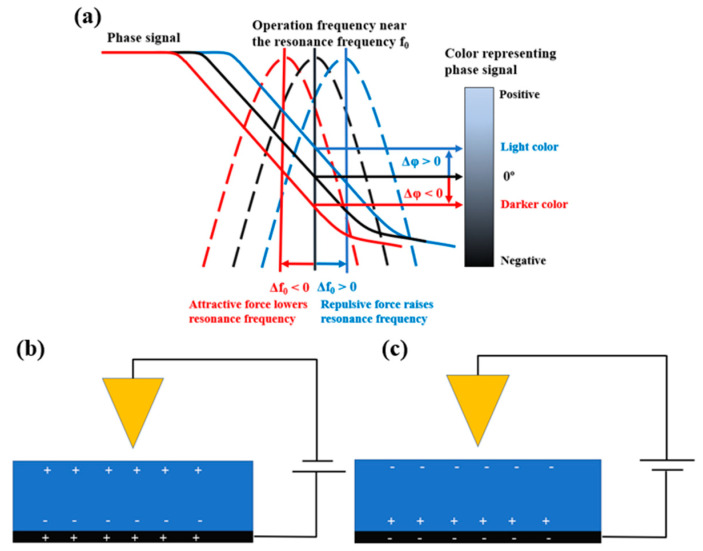
Principle of EFM (**a**) and charge distribution on the membrane surface under positive sample bias voltage (**b**) and negative bias voltage (**c**) [33].

**Figure 2 polymers-15-02295-f002:**
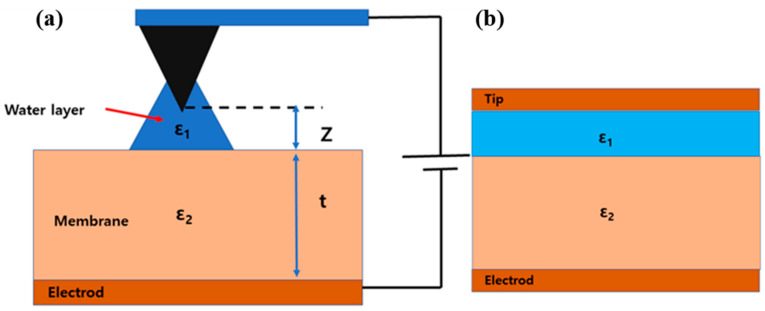
Geometry of a tip and sample surface (**a**) and configuration of a parallel capacitor (**b**).

**Figure 3 polymers-15-02295-f003:**
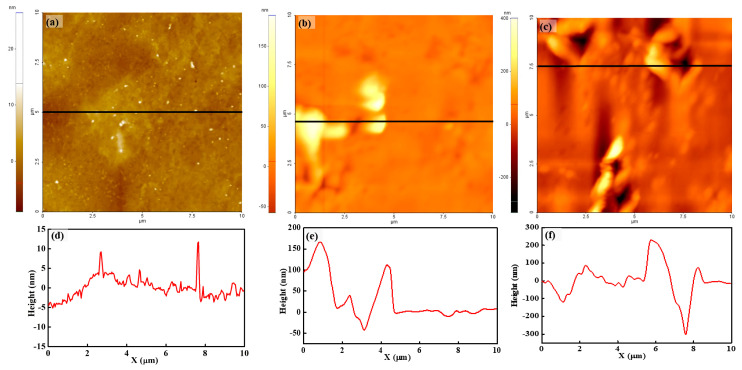
Topography and line profile of recast Nafion (**a**,**d**), the silica–Nafion composite membrane under room humidity (**b**,**e**), and the silica–Nafion composite membrane under a fully hydrated condition (**c**,**f**).

**Figure 4 polymers-15-02295-f004:**
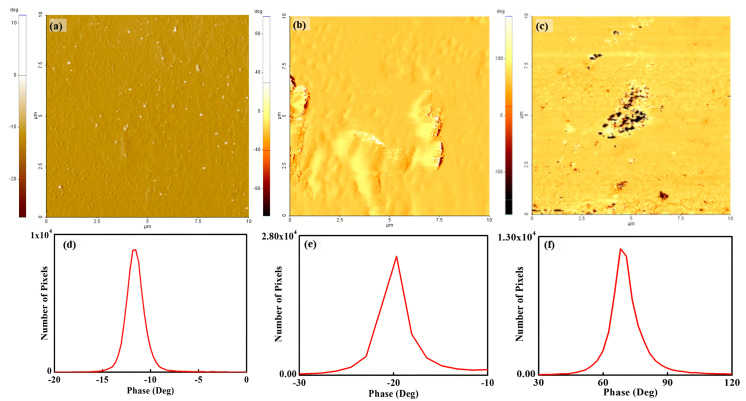
Phase map and histogram of recast Nafion (**a**,**d**), the silica–Nafion composite membrane under room humidity (**b**,**e**), and the silica–Nafion composite membrane under a fully hydrated condition (**c**,**f**).

**Figure 5 polymers-15-02295-f005:**
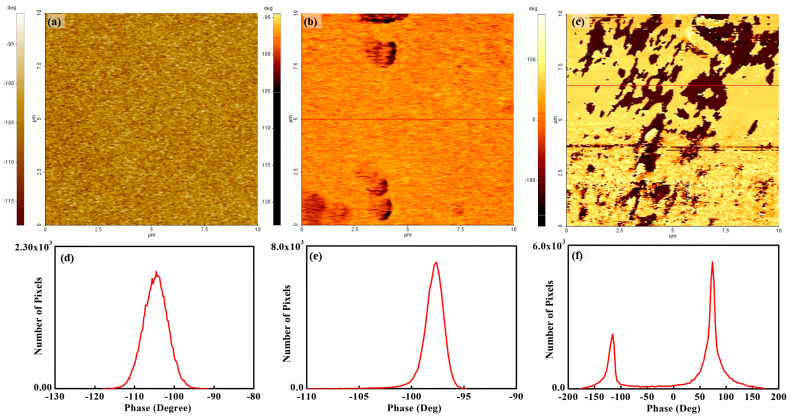
Charge distribution map and histogram of recast Nafion (**a**,**d**), the silica–Nafion composite membrane under room humidity (**b**,**e**), and the silica–Nafion composite membrane under a fully hydrated condition (**c**,**f**) by applying a −2 sample bias voltage.

**Figure 6 polymers-15-02295-f006:**
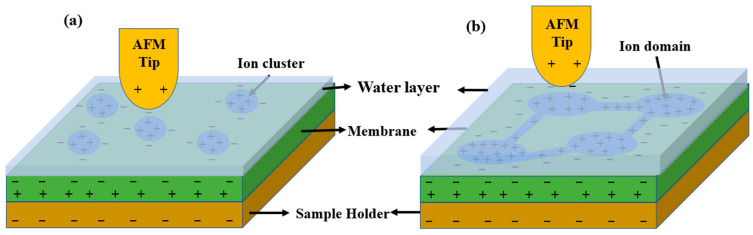
Morphological structure of PEM under different hydration conditions: (**a**) silica–Nafion composite membrane under room humidity condition and (**b**) silica–Nafion composite membrane under a fully hydrated condition.

**Figure 7 polymers-15-02295-f007:**
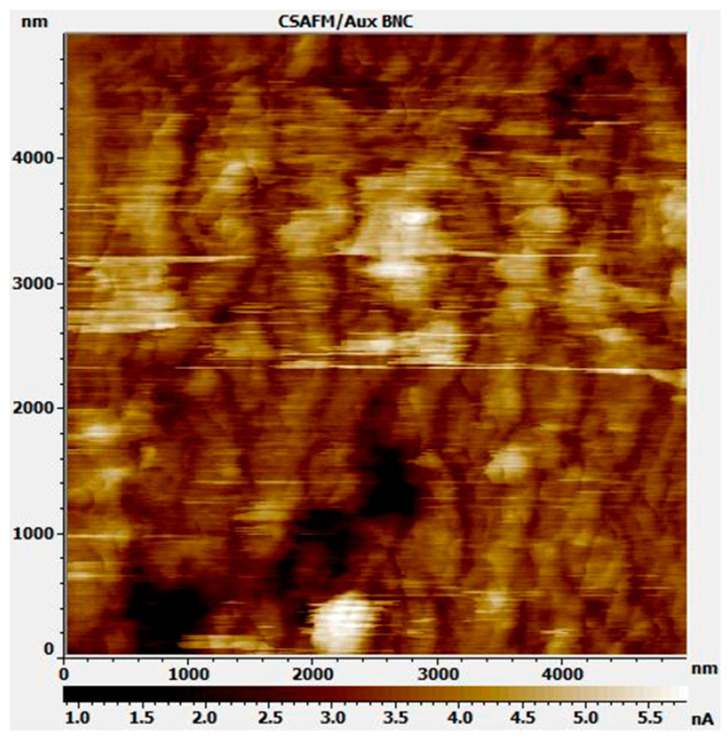
Current distribution image of Nafion 212 using Agilent Pico Scan CSAFM system.

**Table 1 polymers-15-02295-t001:** Conductivity of proton exchange membranes.

	Proton Conductivity (mS cm^−1^)	Max Power Density (mW cm^−1^)	Condition	References
Nafion/SGF-3	3	200	25% RH, 80 °C	[10]
Recast Nafion/SiO_2_/PWA	70		80% RH, 100 °C	[19]
SiO_2_-supported sulfated zirconia/Nafion	70	1000	100% RH, 60 °C	[20]
Nafion composite membranes filled with mesoporous silica	120		95% RH, 80 °C	[21]
Nafion and sulfonated SiO_2_ nanoparticles (6%)	170	850	100% RH, 60 °C	[22]
Multilayer-structured, SiO_2_/sulfonated poly(phenylsulfone) composite membranes	200		100% RH, 80 °C	[23]

**Table 2 polymers-15-02295-t002:** Proton conductivity of recast Nafion and silica–Nafion composite membrane under different RH conditions [34].

	Proton Conductivity (mS cm^−1^)							
	30% RH	40% RH	50% RH	60% RH	70% RH	80% RH	90% RH	100% RH
Recast Nafion	9.5	15	23	33	44	58	80	111
silica–Nafion composite membrane (1 wt.%)	11.5	19	29	42	58	77	104	143

**Table 3 polymers-15-02295-t003:** Parameters and corresponding values for calculating the local dielectric constant.

Parameters	Value
Spring constant, *K*	0.2 N/m
Amplitude, *A*	26 nm
Surface area under the tip, *S*	1 × 10^−14^
Distance between the tip and the membrane surface, *z*	40 nm
Thickness of the membrane, *t*	50 μm
Resonance frequency, *f*_0_	119 kHz
Bias voltage, *V*	2 V

**Table 4 polymers-15-02295-t004:** Dielectric constant of Nafion.

	Dielectric Constant	References
Recast Nafion	14	This study
Nafion 117	23	[37]
Nafion 117	13	[38]
Nafion 117	5~20	[39]

**Table 5 polymers-15-02295-t005:** Parameters and corresponding values for calculating the surface charge of the silica–Nafion composite membrane under room humidity.

Parameters	Value
Phase value of recast Nafion, Δ*Φ*_Nafion_	−104°
Surface area under the tip, *S*	1 × 10^−14^
Amplitude, *A*	26 nm
*Z*	40 nm
*f* _0_	119 kHz
*V*	−2
*d*	50 μm

## Data Availability

Data are contained within the article.

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
