# Peer review of "Quantitative Study of Charge Distribution Variations on Silica–Nafion Composite Membranes under Hydration Using an Approximation Model"

_polymers, 2023, doi:10.3390/polym15102295_

Round 1
Reviewer 1 Report
The manuscript reported quantitative analysis methods for recasting Nafion and silicate–Nafion composite membranes based on the derived mathematical approximation model. It was carried out in several steps: (1) the mathematical approximation model was derived; (2) The phase map and charge distribution map on the PEM were simultaneously derived; (3) The charge distribution maps of the membranes were characterized. It was found that the model was accurately derived and the local dielectric property/surface charge is numerically calculated on the membranes. The calculation result are approximately valid.
I consider the content of this manuscript will definitely meet the reading interests of the readers of the Polymers journal. However, there are certain English spelling and grammar issues, and also the discussion and explanation should be further improved.
Therefore, I suggest giving a minor revision and the authors need to clarify some issues or supply some more experimental data to enrich the content. This could be comprehensive and meaningful work after revision.
Detailed comments can be found in the PDF file.

Author Response
I appreciate the reviewer's feedback on my manuscript. I have revised the manuscript according to their recommendations, with the revised sentences highlighted in red.
- The English of manuscript was edited by professional English editing service.
- I will add recommended key words.
- The sentence was revised by reviewer’s recommend. It is in the page 1 line 22 to 26 of the manuscript.
- The sentence was revised by reviewer’s recommend. It is in the page 2 line 55 of the manuscript.
- The sentence was revised by reviewer’s recommend. It is in the page 2 line 58-61 of the manuscript.
- The sentence was revised by reviewer’s recommend. It is in the page 4 line 171 of the manuscript.
- The sentence was revised by reviewer’s recommend. It is in the page 12 line 454-456 of the manuscript.

Reviewer 2 Report
It is a very interesting report. Characterization of the nanoscale structure of ionomer is a great challenge due to its polymer nature. In this study, the authors proposed an approximation model and successfully applied the model in characterizing the Nafion composite membrane.
A few concerns and suggestions are listed below:
(1) For the experimental design, the author characterized silicate–Nafion composite membrane under both room humidity and fully hydrated condition. However, the Nafion membrane is only analyzed under room humidity conditions. The authors should explain or provide additional characterization.
(2) A brief discussion about why phase shift analysis is important and the principle of how to perform phase shift analysis is suggested.
(3) For equations 1 to 13, please explain each term. For instance, ε0 and Ff need to be explained.
Author Response
I appreciate the reviewer's feedback on my manuscript. I have revised the manuscript according to their recommendations, with the revised sentences highlighted in blue.
(1) The recasting Nafion was analyzed by using quantitatively under dried and fully hydrated condition in previous study. The purpose of using recasting Nafion in this study was used for verifying reliability of the model. The results were compared with other research group’s study. Thus, the Nafion with room humidity data is included in the manuscript. This explanation was added in the manuscript. It is in the page 5 line 210-213.
(2) The importance and brief principal of phase shift analysis were added in the manuscript. It A brief discussion about phase shift analysis is added in the manuscript. It is in the page 3 line 114-121.
(3) Each term is explained. It is in the page 6 line 264-266.

Reviewer 3 Report
1、The innovative points of the manuscript should be more clearly defined.
2、The author should check that the manuscript flows smoothly and conveys a clear meaning, an example being the word "graphene" which appears twice in the sentence from lines 66-68.
3、Check the manuscript, and ensure all the parameters, no matter in equations or in main text, must be formatted in Italic font.
4、The scope of the legend in Figure 3 is not clear enough. In addition, should the scope of the topography and line profile be kept consistent?
5、The tables in the manuscript should use a uniform three-line table format.
6、The figure caption of Figure 6 should be separated from the subsequent analysis.
7、Should the 10 μ in line 531 be 10 μm?
8、The authors should cite more relevant literature to prove the reliability of the model as well as the content of the study.
9.The following literature is suggested to consolidate the presentation.
Lattice Boltzmann simulation of the structural degradation of a gas diffusion layer for a proton exchange membrane fuel cell. Journal of Power Sources, 2023, 556: 232452.
The following literature is suggested to consolidate the presentation.
Lattice Boltzmann simulation of the structural degradation of a gas diffusion layer for a proton exchange membrane fuel cell. Journal of Power Sources, 2023, 556: 232452.
Author Response
I appreciate the reviewer's feedback on my manuscript. I have revised the manuscript according to their recommendations, with the revised sentences highlighted in green.
- The innovative points are added in the introduction. It is in the page 4 line 174 to 181 of the manuscript.
- I tried to fix manuscript flow smoothly and also did English editing by native speaker.
- I checked all parameters again and changed to italic font.
- The legend in Figure 3 is changed to clear enough. I checked consistency of the topography and line profile again and the topography and related line profile is consistent.
- All table formats are changed to uniform three-line table format.
- The caption of Figure 6 was separated. It is in the page 13 line 499 of the manuscript.
- I changed 10 μ x 10 μ to 10 μm x 10 μm.
- I cite relevant literature to prove the reliability of the model. It is in the page 13 table 4 in the manuscript.
- The recommended literature is consolidated in the manuscript. It is in the page 3 line 103 to 111 of the manuscript.

Round 2
Reviewer 2 Report
For the derivation of equation (12) from equation (11), it is better to provide additional information about the relationship between k and d.
Author Response
I appreciate your detailed advice again.
I checked all equation again. I found some mistakes of equations 12 and 13. The letter k is used to dielectric constant and spring constant of a cantilever. That will make confuse of the readers. Thus, I changed the spring constant small letter ‘k’ to capital letter ‘K’ for avoiding confusion. Also, I corrected equation 12 and 13 and related contents in the manuscript. That is highlighted as red color.
I appreciate again for letting me spot the mistake and correct it.
Reviewer 3 Report
I have reviewed the revised manuscript, I think the manuscript can be accepted in the present form.
Author Response
I appreciate your detailed advice again.